# Stage-Dependent Increase of Systemic Immune Activation and CCR5^+^CD4^+^ T Cells in Filarial Driven Lymphedema in Ghana and Tanzania

**DOI:** 10.3390/pathogens12060809

**Published:** 2023-06-07

**Authors:** Abu Abudu Rahamani, Sacha Horn, Manuel Ritter, Anja Feichtner, Jubin Osei-Mensah, Vera Serwaa Opoku, Linda Batsa Debrah, Thomas F. Marandu, Antelmo Haule, Jacklina Mhidze, Abdallah Ngenya, Max Demetrius, Ute Klarmann-Schulz, Michael Hoelscher, Christof Geldmacher, Achim Hoerauf, Akili Kalinga, Alexander Y. Debrah, Inge Kroidl

**Affiliations:** 1Filariasis Unit, Kumasi Centre for Collaborative Research in Tropical Medicine (KCCR), UPO, PMB, Kumasi 00233, Ghana; 2German-West African Centre for Global Health and Pandemic Prevention (G-WAC), Partner Site, UPO, PMB, Kumasi 00233, Ghana; 3Department of Clinical Microbiology, School of Medicine and Dentistry, Kwame Nkrumah University of Science and Technology, UPO, PMB, Kumasi 00233, Ghana; 4Division of Infectious Diseases and Tropical Medicine, University Hospital Munich, Ludwig-Maximilians-Universität (LMU), 80802 Munich, Germany; 5Institute for Medical Microbiology, Immunology and Parasitology (IMMIP), University Hospital Bonn, 53127 Bonn, Germany; 6German Centre for Infection Research (DZIF), Neglected Tropical Diseases, Partner Site, 80802 Munich, Germany; 7Department of Pathobiology, School of Veterinary Medicine, Kwame Nkrumah University of Science and Technology, UPO, PMB, Kumasi 00233, Ghana; 8Mbeya Medical Research Center (MMRC), Department of Immunology, National Institute for Medical Research (NIMR), Mbeya 53107, Tanzania; 9Department of Microbiology and Immunology, Mbeya College of Health and Allied Sciences (UDSM-MCHAS), University of Dar es Salaam, Mbeya 53107, Tanzania; 10National Institute of Medical Research (NIMR)—Headquarters, Dar es Salaam 11101, Tanzania; 11German Centre for Infection Research (DZIF), Neglected Tropical Diseases, Partner Site, Bonn-Cologne, 53127 Bonn, Germany; 12Fraunhofer ITMP, Immunology, Infection and Pandemic Research, 80802 Munich, Germany; 13German-West African Centre for Global Health and Pandemic Prevention (G-WAC), Partner Site, 53127 Bonn, Germany; 14Faculty of Allied Health Sciences, Kwame Nkrumah University of Science and Technology, UPO, PMB, Kumasi 00233, Ghana

**Keywords:** CD4^+^ T cell activation, lymphatic filariasis, lymphedema, CCR5, PD-1, HLADR/CD38, CD8, Tanzania, Ghana

## Abstract

Chronic lymphedema caused by infection of *Wuchereria bancrofti* is a disfiguring disease that leads to physical disability, stigmatization, and reduced quality of life. The edematous changes occur mainly on the lower extremities and can progress over time due to secondary bacterial infections. In this study, we characterized participants with filarial lymphedema from Ghana and Tanzania as having low (stage 1–2), intermediate (stage 3–4), or advanced (stage 5–7) lymphedema to determine CD4^+^ T cell activation patterns and markers associated with immune cell exhaustion. A flow cytometry-based analysis of peripheral whole blood revealed different T cell phenotypes within participants with different stages of filarial lymphedema. In detail, increased frequencies of CD4^+^HLA-DR^+^CD38^+^ T cells were associated with higher stages of filarial lymphedema in patients from Ghana and Tanzania. In addition, significantly increased frequencies of CCR5^+^CD4^+^ T cells were seen in Ghanaian participants with advanced LE stages, which was not observed in the Tanzanian cohort. The frequencies of CD8^+^PD-1^+^ T cells were augmented in individuals with higher stage lymphedema in both countries. These findings show distinct activation and exhaustion patterns in lymphedema patients but reveal that immunological findings differ between West and East African countries.

## 1. Introduction

Lymphatic filariasis, primarily caused by *Wuchereria bancrofti*, is a neglected tropical disease causing substantial public health burdens in filarial endemic areas. While many infected individuals remain asymptomatic, disfiguring pathological changes can develop over time in the form of hydrocele or lymphedema (LE) [1,2]. Filarial lymphedema, along with acute dermatolymphangioadenitis (ADL) attacks, which have been shown to drive inflammation and progression of LE, lead to a decreased quality of life and the stigmatization of individuals [1,3,4,5]. Since the creation of the Global Programme to Eliminate Lymphatic Filariasis (GPELF) in 2000, there have been large mass drug administration (MDA) campaigns globally, including in most African countries, with the aim of disrupting the transmission of LF and thereby halting the spread of the disease [6]. Due to these MDA programs, the number of infected individuals has fallen from approximately 200 million in 1997 to 51 million in 2018 [1]. Tanzania is one such country which has seen large reductions in the prevalence of LF in both the child and adult populations [7]. An additional activity that the GPELF focuses on is morbidity management programs which aim to teach individuals suffering from LE correct hygiene practices to prevent the introduction of bacteria into open wounds, help relieve suffering, and prevent ADL attacks [8]. There are also clinical trials underway which are looking at the effect of doxycycline treatment and the ability to reduce leg LE from either LF or podoconiosis, but as of now, there are no approved medical treatments to prevent the progression of LE [9].

Several studies have shown the central role of T cells in immunity against filariae. Distinct immunological changes have been detected between asymptomatic filarial-infected individuals and others developing LE. While asymptomatic infections are typically characterized by increased regulatory T and B cell responses and a pronounced IgG4 and IL-10 response, chronic LE is associated with pro-inflammatory responses, characterized by elevated pro-inflammatory cytokines such as TNF-α and increased antigen-specific Th1 and Th17 responses in CD4^+^ and CD8^+^ T cells [10,11,12,13,14,15]. Our own previous studies have shown systemic activation on CD4^+^ T cells with increased expression of CD4^+^HLA-DR^+^CD38^+^ T cells in individuals infected with *W. bancrofti* in a cohort in Southwest Tanzania; however, those previous studies did not focus on individuals with chronic LE; therefore, the small numbers of individuals included were not differentiated from those with ongoing helminth infections [16]. Moreover, our group has previously found an altered expression of exhausted CD8^+^ and CD4^+^ T cells within a filarial LE cohort compared to asymptomatic *W. bancrofti*-infected individuals, as seen by the change in expression of receptors, including programmed cell death-1 (PD-1), killer cell lectin-receptor subfamily G member 1 (KLRG-1), lymphocyte activation gene 3 (LAG-3), and T-cell immunoglobulin and mucin-domain containing-3 (Tim-3) [17,18]. These previous studies only included limited numbers of lymphedema participants, making it difficult to evaluate any immunological differences between the LE stages.

Thus, in this study, we compare large cohorts of participants with varying stages of LE from two African countries, Ghana and Tanzania. The frequencies of CD4^+^ T cell activation and markers associated with exhaustion are measured to characterize how LE progression affects the phenotypic expression of CD4^+^ and CD8^+^ T cells.

## 2. Materials and Methods

### 2.1. Study Population and Parasitic Assessment

Nighttime blood collections were performed to draw venous whole blood in 9 mL EDTA tubes (Sarstedt, Nümbrecht, Germany) from participants in the Upper East Region of Ghana (Navrongo, Kassena-Nankana East Municipal District) and Southeast Tanzania (Lindi Region) between 2018 and 2020 as part of the TAKeOFF LEDoxy clinical trial (01KA1612). At the time of sample collection, clinical evaluations were performed on each participant along with an epidemiological-based survey. The questionnaire included questions regarding gender, age, number of years lived in the filarial endemic area, if they had ever received ivermectin (IVM) and albendazole (ALB) during MDA campaigns and if so, how many times IVM and ALB had been taken, history of ever having experienced an ADL attack, and information regarding occupation. Participants with HIV, fever, or malaria at the time of sampling were omitted from analysis and directed to the nearest clinic for further diagnosis and treatment. At the time of recruitment, participants were excluded if they were less than 14 or over 65 years of age, had a body weight below 40 kg, had evidence of severe comorbidity other than filariasis, or had a medical history or intake of drugs which would interact with the study drug doxycycline within the LEDoxy clinical trial (trial numbers ISRCTN65756724 and ISRCTN14042737).

In addition to the clinical examination, the Filariasis Test Strip (FTS; previously Alere, now Abbott Laboratories, Chicago, IL, USA) and TropBio Og4C3 Filariasis Antigen ELISA (hereafter referred to as TropBio, Cellabs, Brookvale, Australia) were used to test the blood of all participants for the presence of circulating filarial antigen. Based on the results of these tests, participants were then categorized as uninfected control individuals, *W. bancrofti*-infected (Wb-infected), or having filarial-driven leg lymphedema (LE). The uninfected control cohort was defined as individuals who had been living in the filarial endemic area for at least 5 years, tested negative for both FTS and TropBio tests, and had no visible signs of LE. The Wb-infected individuals all tested positive for the FTS and TropBio tests and displayed no signs of LE. For participants with LE, the Dreyer lymphedema staging protocol [19] was used to define the LE group. Individuals with LE were allocated into low (stage 1–2), medium (stage 3–4), or advanced (stages 5–7) LE stage groups based on the level of pathology they displayed. Table 1 displays the descriptive characteristics of the first cohort of participants examined, including uninfected control (*n* = 34), *W. bancrofti*-infected (*n* = 10), and filarial LE (*n* = 25) cohorts from Ghana. Recruitment of additional LE individuals was then undertaken in Ghana (*n* = 60) and extended to Tanzania (*n* = 84), and a supplementary flow cytometry panel was utilized to examine additional markers; characteristics from these additional cohorts can be found in Table 2.

### 2.2. Ethics

All samples were taken as part of the ongoing German Federal Ministry of Education and Research (BMBF)-funded TAKeOFF LEDoxy clinical trial or the German Research Foundation (DFG)-funded RHINO project. Participants were included if they provided written informed consent and were over 14 years of age. Ethical approval was obtained from the Ethics Committee of the LMU Munich, Germany (17-858, LEDoxy and 18-377, RHINO), the Ethics Committee at the University Hospital of Bonn, Germany (359/17, LEDoxy and 041/18, RHINO), the Committee on Human Research Publication and Ethics at the Kwame Nkrumah University of Science and Technology in Kumasi, Ghana (CHRPE/AP/525/17, LEDoxy and CHRPE/AP/235/18, RHINO), The Ghana Food and Drugs Authority (FDA/CT/181 and FDA/CT/181(1)), the Ghana Health Services (GHS-ERC-007/07/17), and the Medical Research Coordinating Committee (MRCC) at the National Institute for Medical Research (NIMR) in Dar es Salaam, Tanzania (NIMR/HQ/R.8a/Vol.IX/2693, TFDA0017/CTR/0020/3, LEDoxy and GB.152/377/01/194, RHINO).

### 2.3. Flow Cytometric Analysis

Peripheral whole blood samples were processed in both Ghana and Tanzania according to the previously described protocol [20]. In brief, peripheral whole blood was collected in a sodium heparin blood collection tube (Sarstedt, Nümbrecht, Germany) and 100 µL per panel were removed for further processing. First, whole blood was incubated with the following extracellular anti-human antibodies for 30 min at room temperature: CD4-PerCp-Cy5.5 (Invitrogen, Carlsbad, USA, clone OKT4), HLA-DR-PeCy7 (Invitrogen, Carlsbad, USA, clone LN3), and CD38-APC (Biolegend, San Diego, CA, USA, clone HIT2). Following the incubation, cells were lysed with 1× BD FACS^TM^ lysis buffer (BD^TM^ Biosciences) for 10 min before centrifugation (Hettich Rotina 420R, Tuttlingen, Germany) for 5 min at 600× *g*. Cells were subsequently resuspended in a pre-chilled freezing solution (inactivated Fetal Bovine Serum with 10% DMSO) and transferred to a Nunc™ CryoTube™ (Thermo Fisher Scientific, Waltham, MA, USA). The tubes were first frozen at −20 °C overnight in a StrataCooler Preservation module (Agilent Technologies, Santa Clara, CA, USA) and, afterwards, were stored long-term in liquid nitrogen before they were transferred to Germany. Upon transfer, cells were thawed in a 37 °C water bath and subsequently washed two times with 2.5 mL thawing media (RPMI 1640 media GlutaMAX supplement (Invitrogen, Carlsbad, CA, USA) with 10% FCS (Sigma-Aldrich, St. Louis, MO, USA), 1% penicillin-streptomycin (10,000 U/mL, Sigma-Aldrich), and 0.2 µL/mL Benzonase^®^ Nuclease (25 U/µL, Merck Millipore, Kenilworth, NJ, USA)) and centrifuged at 400× *g* for 5 min. Upon centrifugation, 37.5 µL FCS (Sigma-Aldrich, Munich, Germany) were added to the cells, followed by 1 mL eBioscience^TM^ Fixation/Permeabilization Concentrate and Diluent solution (diluted 1:4, Invitrogen, Carlsbad, CA, USA) and a 25 min incubation at 4 °C. Cells were then centrifuged again at 600× *g* for 5 min, followed by washing with 2mL of 1× eBioscience^TM^ Permeabilization Buffer (Invitrogen, Carlsbad, CA, USA) and an additional 600× *g* for 5 min centrifugation. Next, the following intracellular antibodies were added: CD3-ECD (Beckman Coulter, Brea, CA, USA, clone UCHT1) and PD-1-APC-AF700 (Biolegend San Diego, CA, USA, HIT2). Post incubation, cells were again washed with the aforementioned 1× Permeabilization Buffer and centrifuged at 600× *g* for 5 min before the addition of 200 µL of 1× BD CellFIX (BD™ Biosciences). A 13-channel CytoFlex S flow cytometer (Beckman Coulter) was then used to acquire the cells, which were subsequently analyzed using FlowJo_v10.6.0 software (FlowJo LLC, Ashland, OR, USA). The applied gating strategy (Appendix A) was created using fluorescence minus one controls, as previously published [20]. The VersaComp Antibody Capture Kit (Beckman Coulter) was used to perform compensation.

### 2.4. Statistical Analysis

GraphPad Prism version 6.01 (GraphPad Software, Inc., La Jolla, CA, USA) was used to compare each of the parameters between groups. Variables showed non-parametric distribution according to the Kolmogorov–Smirnov test, and, therefore, the Kruskal–Wallis test was used to test differences between the overall groups. If significant (*p* < 0.05), further comparison was carried out with the Dunn’s post hoc test. Spearman correlation was used for comparison of continuous parameters. CRAN R 3.6.2 was then used to determine if there was any influence from potential confounders on the elevated CD4^+^ and CD8^+^ T cell marker frequencies. We performed uni- and multi-variable linear regression analysis and adjusted for any potential confounders, which were selected with a stepwise regression model (site, gender, age, years living in the endemic areas, number of years with lymphedema, number of previous MDA treatment rounds, and stage group (low, medium, or advanced) of lymphedema).

## 3. Results

### 3.1. Increased Systemic CD4^+^ T Cell Activation among Individuals with Leg Lymphedema as Compared to Other Groups Residing in Filarial Endemic Areas

Overall, samples from 188 participants were analyzed, with 144 having filarial leg lymphedema. Initially, 70 Ghanaian participants from the Upper East Region of Ghana were recruited. The systemic immune activation levels of the individuals with LE residing in filarial endemic areas in the Upper East Region of Ghana (*n* = 25) were compared with uninfected control (*n* = 34) and individuals who were infected with *W. bancrofti* but did not display signs of LE (*n* = 10) (Table 1). The frequencies of CD3^+^ and CD4^+^ T cells between the groups were analyzed and showed no significant difference between the three cohorts (Figure 1A,B). In contrast, elevated frequencies of activated CD4^+^HLA-DR^+^CD38^+^ T cells were seen in the LE group when compared to the control and asymptomatic infection group (median values of 5.56%, 5.55%, and 7.13%, *p* = 0.0166 in uninfected control., *W. bancrofti*-infected, and LE, respectively) (Figure 1C). All of the samples were gated according to the whole blood gating strategy presented in Appendix A.

### 3.2. Increased Systemic Immune Activation and CCR5 Expression Are Associated with Advanced LE Stage in CD4^+^ T Cells

Since there were differences in the levels of systemic immune activation present within the LE cohort in Ghana, additional LE participants were recruited from Ghana (*n* = 60) along with cohorts from Tanzania (*n* = 84) to further examine their systemic immune activation and, additionally, exhaustion changes present within the various LE stage groups. In Ghana, there were 25 participants in the cohort with low pathology (Dreyer stages 1–2), 15 participants with medium pathology (Dreyer stages 3–4), and 20 individuals in the cohort with advanced stages of lymphedema (Dreyer stages 5–7) [19]. The LE cohorts in Tanzania consisted of lymphedema participants with low (*n* = 24), medium (*n* = 36), and advanced (*n* = 24) stages (Table 2). Information regarding the number of ADL attacks per cohort can be found in Appendix A. Of note, 49% of the participants from Tanzania reported no ADL attacks during the year before recruitment compared with only 13% in Ghana. In addition, the time since LE onset was much longer for most Tanzanian patients (26 years and 14 years in Tanzania and Ghana, respectively) and the LE cohort from Tanzania included more men than the Ghanaian cohort (41.7% and 11.7% in Tanzania and Ghana, respectively). Other variables such as age, time spent in the endemic area, and number of previous MDA rounds were comparable between the cohorts in both countries.

The LE stage groups from both countries differed regarding the overall frequencies of CD3^+^ and CD4^+^ T cells. In general, participants from Ghana had overall significantly higher CD3^+^ and CD4^+^ T cell frequencies than the participants from Tanzania with similar frequencies of CD8^+^ T cells (Appendix A). Despite the lower CD4^+^ T cell frequencies, the Tanzanian participants displayed higher levels of CD4^+^HLA-DR^+^CD38^+^ activated T cells when compared to those from Ghana (median of 2.46% and 6.19%, *p* < 0.0001 in the low stages group, 2.77% and 6.14%, *p* < 0.0001 in the medium stages group, and 3.56% and 9.02%, *p* < 0.0001 in the advanced stages group from Ghana and Tanzania, respectively). The T cell counts are comparable between the CD3, CD4, and CD8 T cell groups and are shown in Appendix A.

In Ghana, the frequencies of CD3^+^ T cells were found to be similar across all of the LE stage groups (Figure 2A); however, there was a significant decrease in the level of CD4^+^ T cells seen between the advanced LE stages and both the low and medium stages (Figure 2B). A weak negative correlation was found between the LE stage and the CD4^+^ T cell frequencies (*p* = 0.0296, rho = −0.2812, Appendix A). In contrast, the frequency of CD4^+^HLA-DR^+^CD38^+^ T cells was significantly increased in the advanced stages of LE (median of 2.46%, 2.77%, and 3.56% for low, medium, and advanced, respectively) (Figure 2C) and the correlation analysis confirmed this weak positive correlation (*p* = 0.0321, rho = 0.2771, Appendix A). Similarly, there were also statistically significant elevated frequencies of CD4^+^CCR5^+^ T cells in the advanced stages when compared to the low stages of LE (Figure 2D, *p* = 0.0097), and a similar trend was observed in the CD4^+^PD-1^+^ T cells (Figure 2E). The correlation analysis revealed a weak positive correlation between the LE stage and CD4^+^CCR5^+^ T cell frequencies (*p* = 0.0082, rho = 0.3892). While not statistically significant, we observed a noticeable tendency towards higher CD8^+^ (Figure 3A), CD8^+^CCR5^+^ (Figure 3B), and CD8^+^PD-1^+^ (Figure 3C) T cell frequencies in the Ghanaian LE individuals with more advanced LE stages, when compared to the low and medium groups.

Regardless of the leg LE stage, comparable frequencies of CD3^+^ and CD4^+^ T cells were observed when we examined samples taken from the Lindi region in Tanzania (Figure 4A,B). Statistically significant elevated frequencies of CD4^+^HLA-DR^+^CD38^+^ T cells were detected in the more advanced stages of LE (stages 5–7) as compared to the low and medium stages in the Tanzanian participant samples (Figure 4C, *p* = 0.0141), and these results are supported by the correlation between LE stages and the activation markers (*p* = 0.0093, rho = 0.2807, Appendix A). Interestingly, we did not observe the same increasing tendency in the CD4^+^CCR5^+^ frequencies associated with increased LE stages in Tanzania that were seen in Ghana (Figure 4D). When examining the CD4^+^PD-1^+^ populations, there were no significant differences between the groups (Figure 4E). In the Tanzanian cohort, there was relatively similar expression of CD8^+^ T cells (Figure 5A) across the groups, while CD8^+^CCR5^+^ expression had a tendency to increase with the advanced stages of LE when compared to the medium stages (Figure 5B). The characteristically exhausted CD8^+^PD-1^+^ T cells showed elevated expression in the advanced group when compared to the medium stages, along with an increased tendency in comparison to the low stages (Figure 5C, *p* = 0.0455 and *p* = 0.0008 compared to low and medium groups, respectively).

In order to address whether the increase in systemic immune activation, CCR5^+^, and PD-1^+^ cells was due to confounding factors, we performed uni- and multi-variable regression analysis. There was a significant difference between Ghana and Tanzania, where the cell frequencies were lower in Tanzania than in Ghana (coef. −8.83, *p*-value < 0.001) (Table 3). Univariable analysis for the risk factors after adjusting for the site effect revealed that there was an influence of gender (coef. −5.59, *p*-value 0.001) and advanced LE stage (coef. −4.70, *p*-value 0.01) on the CD4^+^ T cell frequencies. However, the mutually adjusted multivariable regression model did not show any of the risk factors to be significant. The same analysis was then carried out with the CD4^+^HLA-DR^+^CD38^+^ T cell frequencies, again adjusting for site differences between Ghana and Tanzania (coef. 4.28, *p*-value < 0.001) (Table 4). Univariable analysis for the risk factors after adjusting for the site effect revealed an influence of age and advanced LE stage. The multivariable regression analysis revealed that, even after adjustment, the advanced LE stage still showed an influence on CD4^+^HLA-DR^+^CD38^+^ T cell frequencies (coef. 3.30, *p*-value < 0.001). When examining the effect of risk factors on CD4^+^CCR5^+^ expression, we saw no influence of any of them in the univariable analysis; however, in the adjusted multivariable regression model, we observed a positive association of the number of years an individual has been living with LE to CD4^+^CCR5^+^ frequency (coef. 0.003, *p*-value 0.03) (Table 5). A similar analysis was, again, done on the CD8^+^PD-1^+^ expression, and it was found that there was a site effect and advanced LE stage present in the univariable analysis, both of which were negated during the multivariable analysis adjustment (Table 6).

## 4. Discussion

Treatment campaigns targeting lymphatic filariasis over the last 20 years have been very successful in reducing the number of infected individuals by 75% through lowering *W. bancrofti* transmission; however, there now remain large numbers of individuals suffering from chronic filarial LE and the associated pathological changes, such as chronic LE and hydrocele [1]. Modulation of the immune system during *W. bancrofti* infections is characterized by alternatively activated macrophages, the increased systemic immune activation of CD4^+^ cells, changes in the regulatory T and B cells, and an altered Th2 response, while those with filarial LE have been seen to display increased pro-inflammatory responses and an altered exhausted CD8^+^ T cell profile [12,16,17,21,22,23,24].

In this study, a small subset of filarial LE participants from Ghana (*n* = 25) was initially sampled and compared with uninfected control (*n* = 34) and Wb-infected (*n* = 10) individuals. This same participant cohort was recently seen to display distinct patterns of exhausted effector and memory CD4^+^ and CD8^+^ T cell subsets and, thus, we were interested in determining if they also displayed altered activation profiles, since T cell exhaustion is often caused by immune activation, persistent antigen stimulation, and inflammation [17,18,25,26]. Building on our previous analysis, we conducted a more in-depth evaluation of immune activation patterns in filarial lymphedema participants with different stages by including more participants from two different study sites; we showed that there is indeed an increase in systemic CD4^+^ T cell activation associated with the progression of chronic LE within a relatively large cohort of LE participants. A previous study by our group found an association between *W. bancrofti* infection in general and T cell activation [16]. However, in the previous study, due to low numbers of LE participants, the LE group could not be evaluated independently from the *W. bancrofti*-infected group, leaving a gap in understanding of how systemic immune activation differs in LE pathology. In this study, the opposite limitation occurred in that there were very few *W. bancrofti*-infected individuals (*n* = 10) found in these heavily MDA-treated areas in Ghana, and the individuals who were found had low worm intensity [27]. This could potentially be why there were no significant differences found between the systemic CD4^+^ T cell activation levels of the *W. bancrofti*-infected and LE groups (Figure 1).

The presence of dead worms in the lymphatic system can trigger exacerbated immune responses and inflammation, leading to the clearance of infection and lymphedema [4]. Other groups have found it valuable to study chemokine receptors in relation to lymphatic filariasis pathology, as seen in the work by Babu et al., 2005; although there was a small number of participants in the chronic LE group, they demonstrated a significantly higher percentage of CCR9-expressing T cells in the LE group [28]. CCR5 is a chemokine that has classically been known as a common co-receptor for HIV on CD4^+^ T cells, but it also plays a role in other bacterial, viral, and parasitic infections [29,30,31,32,33,34,35]. While there have been few studies examining the profile of CCR5 and other chemokines in relation to filarial LE, our group previously examined the frequencies of CCR5 and an array of memory markers among a Ghanaian cohort of individuals that suffer from LE in comparison to other non-LE individuals residing in filarial endemic areas [17,18]. We saw evidence of higher frequencies of CD8^+^CCR5^+^ per se, as well as increased frequencies of CCR5^+^CD45RA^−^ memory T cells among the LE group. While expecting to observe a similar trend in this study with a larger cohort of LE individuals, we instead observed an increase only in the CD4^+^CCR5^+^ T cells and an increasing trend for CD8^+^CCR5^+^ T cells associated with an increasing LE stage in the Ghanaian cohort and with minute differences seen between the LE stage groups in the Tanzanian cohort. However, these differences were shown to be negated by outside factors when adjusted for in the multivariable regression analysis. Given the nature of the unclear patterns which we observed in the CD4^+^ and CD8^+^ T cell populations expressing CCR5, and along with what other groups have seen regarding the chemokine CCR9 in chronic LE, there are indications that chemokines play a complex role in the host immune response in the development of filarial LE [28].

Another marker which was seen to be altered during chronic LE is programmed cell death protein 1 (PD-1); during inflammation, PD-1 plays an immunoregulatory role, acts as an immune barrier molecule to down-regulate T cell activities, and helps to control the T cell response and maintain peripheral tolerance [17,18,36,37]. To better characterize important markers related to T cell exhaustion in LE, our group previously defined the co-expression of PD-1, IFN-γ, and IL-10 on exhausted memory and effector populations in a subset of Ghanaian LE individuals [17,18]. We chose, in the current study, to focus primarily on PD-1 to characterize exhausted T cells in a larger cohort of LE individuals. While we saw minimal changes in the frequencies of CD4^+^PD-1^+^ cells across the varying LE groups, we did observe a trend towards increasing CD8^+^PD-1^+^ cells in the Ghanaian cohort of advanced LE stages and a significant increase in the Tanzanian advanced LE stages group. The broad range of PD-1 expression in CD4^+^ T cells in the Tanzanian samples was striking, since the same spread was not observed in the Ghanaian samples or CD8^+^PD-1^+^ T cells. We, again, observed, in the multivariable regression, that the effect of accounting for the covariates explained the changes which we had previously observed. Of note, we observed that there are indeed higher frequencies of CD3^+^ and CD4^+^ T cells in the Ghanaian cohort as compared to the Tanzanian cohort (Appendix A). Despite the lower CD4^+^ T cell frequencies, there were even higher levels of systemic immune activation markers within the Tanzanian population.

Our analysis revealed some notable differences between the two cohorts. There was a higher percentage of women with LE and a higher percentage of participants reported experiencing an ADL attack within the last year in Ghana. The time since the onset of LE was considerably longer in Tanzania across all stage groups. Because Ghana and Tanzania are located in West and East Africa, respectively, we expect that there are underlying differences in exposure to environmental factors, a variety of infectious diseases, and ethnic backgrounds that could play a role in shaping the immune response to *W. bancrofti* [38,39,40].

Filarial parasite *W. bancrofti* infection is known to be associated with key pathological conditions including hydrocele, lymphedema, and elephantiasis [28,41]. The processes involved in the development of filarial pathology are complex and involve extrinsic factors, including secondary bacterial infections, and intrinsic factors such as the host immune response [19,42]. Filarial lymphedema patients, especially those suffering from more advanced stages of LE, have long been known to be at a higher risk of bacterial infections and ADL attacks [43,44]. The more advanced stages of LE are associated with skin folds, entry lesions, and fissures in the skin, which can facilitate bacterial entry through the skin [5,19,45]. Not only can bacteria gain easier entry but, once in the body, bacteria are allowed to grow uninhibited through stagnation in the lymphatics [44]. This in turn leads to an increase in ADL attacks, and in a cyclical pattern, the ADL also tends to worsen the lymphedema [3,43,45]. This was also observed within our cohorts; along with increasing pathology comes an increased number of ADL attacks suffered within the last year (Appendix A). Many of those with more advanced stages of LE and a high number of ADL attacks also experience a decreased quality of life, increased disability-adjusted life years, social stigma, and poverty [46,47,48,49]. Not only do those with a more advanced LE stage have a greater chance for bacteria to enter through open wounds and suffer from more ADL attacks, but we also show here for the first time that these individuals display higher levels of systemic T cell activation at the same time. Whether this increase in systemic activation is caused by bacteria, the inflammation caused by the LE, or the immune response due to persistent antigen is yet to be shown and could be the focus of future research. It is also important for national NTD programs operating in sub-Saharan Africa to further consider immune profile differences between inhabitants of filarial endemic areas and, most importantly, how these differences affect susceptibility to disease and MDA response.

There were a number of limitations of this study which are important to recognize. While the flow cytometry field method that we utilized produces high quality results, the extracellular markers must be stained in the field prior to freezing. They cannot be changed afterwards to address any new research questions and, therefore, potentially limit the possible analysis and applications. We also recognize that careful interpretation is needed due to the fact that we were not given the opportunity to test for other endemic parasitic infections (e.g., Kato–Katz for soil-transmitted helminths) at the time of recruitment. Be that as it may, it can be assumed that there are even distributions of any other infections across all of our groups and, thus, they would not affect our outcomes.

## 5. Conclusions

We examined the systemic immune activation, CCR5, and PD-1 characteristics of Ghanaian and Tanzanian individuals suffering from various LE stages, ranging from low (stages 1–2) through advanced stages (stages 5–7). With this relatively large cohort of such individuals from two filarial endemic regions in Africa, we observed that filarial LE individuals in both countries displayed similar patterns of systemic immune activation, with more advanced LE stages showing increased frequencies of the activation markers HLA-DR and CD38 on CD4^+^ T cells. While there was a trend seen towards increasing frequencies of CCR5^+^CD4^+^ and PD-1^+^CD8^+^ T cells, further analysis would need to be performed to confirm a clear pattern within filarial infection. We can conclude that, at this point in time, future studies and clinical trials need to take into consideration this increase in systemic activation when designing national ADL attack prevention and morbidity control strategies for LE patients.

## Figures and Tables

**Figure 1 pathogens-12-00809-f001:**
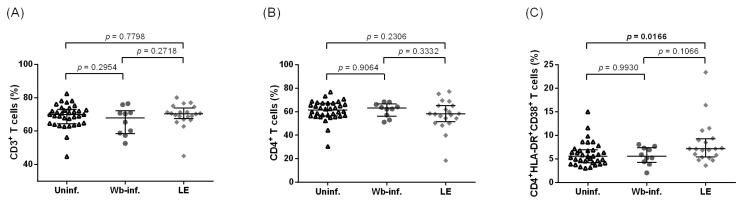
Increased systemic immune activation among leg lymphedema individuals in filarial endemic areas. Cell populations were analyzed according to the applied gating strategy (Appendix A). Multiparametric flow cytometry analysis was used to measure the frequencies of CD3^+^ T cells (**A**), CD4^+^ T cells (**B**), and CD4^+^HLA-DR^+^CD38^+^ T cells (**C**) on individuals from filarial endemic areas of the Upper East Region of Ghana. Individuals were classified as uninfected controls (Uninf., *n* = 34), *Wuchereria bancrofti*-infected (Wb-inf., *n* = 10), or presenting leg lymphedema pathology (LE, *n* = 26). Kruskal–Wallis followed by Dunn’s multiple comparison post hoc analysis was used to find statistical significances between the groups (*p* < 0.05 statistically significant and shown in bold).

**Figure 2 pathogens-12-00809-f002:**
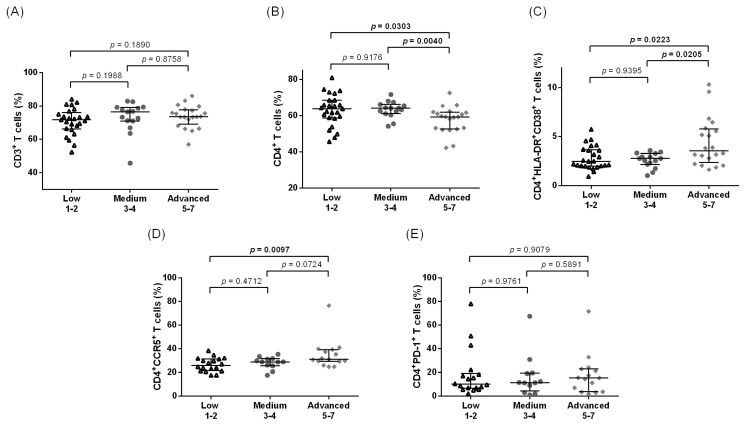
Frequencies of CD3^+^, CD4^+^, and CD4^+^ T cell populations among Ghanaian lymphedema individuals. Cell populations were analyzed according to the applied gating strategy (Appendix A). Multiparametric flow cytometry analysis was used to measure the frequencies of CD3^+^ (**A**), CD4^+^ (**B**), CD4^+^HLA-DR^+^CD38^+^ (**C**), CD4^+^CCR5^+^ (**D**), and CD4^+^PD-1^+^ T cells (**E**) in individuals displaying chronic leg lymphedema from *Wuchereria bancrofti* infection. All individuals were classified according to the Dreyer lymphedema scale [19] and classified as having low (stages 1–2, *n* = 25), medium (stages 3–4, *n* = 15), or advanced grade (stages 5–7, *n* = 20) leg lymphedema. Kruskal–Wallis followed by Dunn’s multiple comparison post hoc analysis was used to find statistical significances between the groups (*p* < 0.05 statistically significant and shown in bold).

**Figure 3 pathogens-12-00809-f003:**
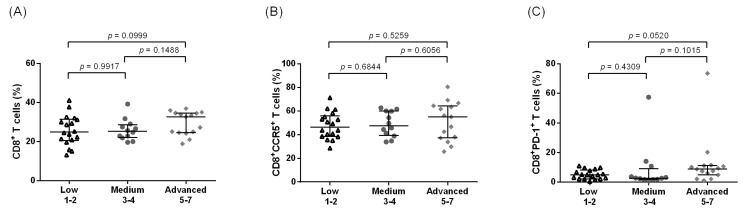
Frequencies of CD8^+^, CD8^+^CCR5^+^, and CD8^+^PD-1^+^T cell populations among Ghanaian lymphedema individuals. Cell populations were analyzed according to the applied gating strategy (Appendix A). Multiparametric flow cytometry analysis was used to measure the frequencies of CD8^+^ (**A**), CD8^+^CCR5^+^ T cells (**B**), and CD8^+^PD-1^+^ T cells (**C**) in individuals displaying chronic leg lymphedema from *Wuchereria bancrofti* infection. All individuals were classified according to the Dreyer lymphedema scale [19] and classified as having low (stages 1–2, *n* = 18), medium (stages 3–4, *n* = 12), or advanced grade (stages 5–7, *n* = 15) leg lymphedema. Kruskal–Wallis followed by Dunn’s multiple comparison post hoc analysis was used to find statistical significances between the groups (*p* < 0.05).

**Figure 4 pathogens-12-00809-f004:**
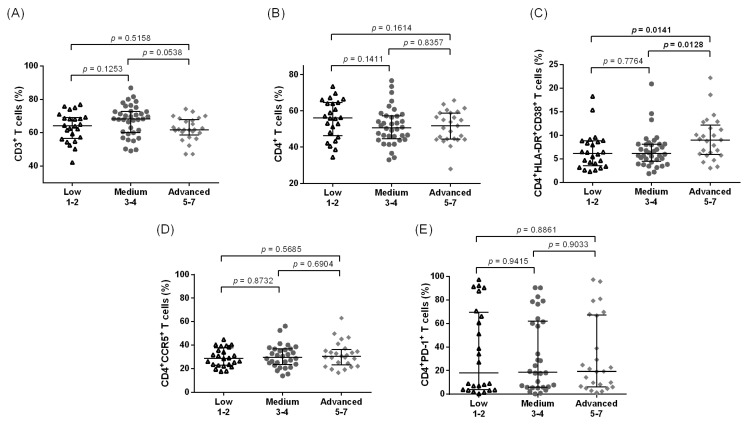
Increased frequencies of systemic CD4^+^ T cell activation of Tanzanian leg lymphedema individuals. Cell populations were analyzed according to the applied gating strategy (Appendix A). Multiparametric flow cytometry analysis was used to measure the frequencies of CD3^+^ (**A**), CD4^+^ (**B**), CD4^+^HLA-DR^+^CD38^+^ (**C**), CD4^+^CCR5^+^ (**D**), and CD4^+^PD-1^+^ T cells (**E**) in individuals displaying chronic leg lymphedema from *Wuchereria bancrofti* infection. All individuals were classified according to the Dreyer lymphedema scale [19] and classified as having low (stages 1–2, *n* = 24), medium (stages 3–4, *n* = 36), or advanced grade (stages 5–7, *n* = 24) leg lymphedema. Kruskal–Wallis followed by Dunn’s multiple comparison post hoc analysis was used to find statistical significances between the groups (*p* < 0.05 statistically significant and shown in bold).

**Figure 5 pathogens-12-00809-f005:**
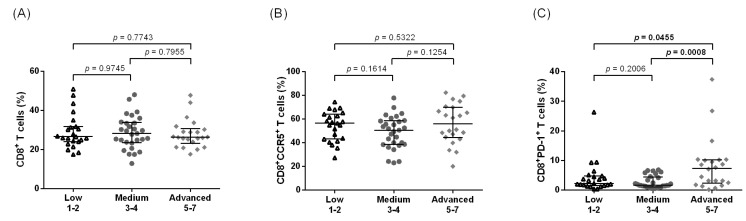
Frequencies of CD8^+^, CD8^+^CCR5^+^, and CD8^+^PD-1^+^ T cell populations among Tanzanian leg lymphedema individuals. Cell populations were analyzed according to the applied gating strategy (Appendix A). Multiparametric flow cytometry analysis was used to measure the frequencies of CD8^+^ (**A**), CD8^+^CCR5^+^ T cells (**B**), and CD8^+^PD-1^+^ T cells (**C**) in individuals displaying chronic leg lymphedema from *Wuchereria bancrofti* infection. All individuals were classified according to the Dreyer lymphedema scale [19] and classified as having low (stages 1–2, *n* = 23), medium (stages 3–4, *n* = 29), or advanced grade (stages 5–7, *n* = 22) leg lymphedema. Kruskal–Wallis followed by Dunn’s multiple comparison post hoc analysis was used to find statistical significances between the groups (*p* < 0.05 statistically significant and shown in bold).

**Table 1 pathogens-12-00809-t001:** Population characteristics. Study participants were classified as endemic normal (Uninf.), *Wuchereria bancrofti*-infected (Wb.inf.), or lymphedema (LE) based on pathology, according to Dreyer staging [19]. The table shows the sample size (*n*), median age and weight, gender, mean years living in the filarial endemic areas, median MDA received, how groups were characterized (FTS and TropBio test results, respectively), history of ever experiencing ADL attacks, and occupation.

	Uninf.	Wb-inf.	LE
*n*	34	10	25
Age, median (range)	50 (30–81)	45.50 (31–83)	47.50 (32–64)
Females, *n* (%)	21 (61.8)	6 (60)	22 (88)
Mean years living in the endemic area (range)	41.76 (22–81)	48 (28–83)	49.31 (32–64)
Median MDA rounds (range)	1 (1–6)	3.50 (1–5)	6 (1–10)
FTS results/TropBio result	−/−	+/+	−/−
History of ADL attacks, *n* (%)	NA	NA	25 (100)
Occupation, *n* (%)			
Farmer	NA	NA	18 (72)
Trader	NA	NA	7 (28)
Weight, median (range)	NA	NA	65.5 (53.5–100)

**Table 2 pathogens-12-00809-t002:** Lymphedema participant characteristics by stage. Lymphedema (LE) study participants were classified as having low (stages 1–2), medium (stages 3–4), or advanced (stages 5–7) LE. All LE classifications were made according to the Dreyer scale [19]. Table shows the sample size (*n*), median age and weight, gender, mean years living in the filarial endemic areas, median MDA rounds received, if they had a history of ADL attacks, and occupation.

	Ghana				Tanzania			
	Total	Low	Medium	Advanced	Total	Low	Medium	Advanced
*n*	60	25	15	20	84	24	36	24
Age, median (range)	47.50 (19–65)	47 (26–65)	47 (26–62)	53 (19–64)	48.5 (23–65)	51.5 (23–65)	51 (23–65)	51 (23–65)
Females, *n* (%)	53 (88.3)	21 (84)	15 (100)	17 (85)	49 (58.3)	16 (66.7)	22 (61.1)	11 (45.8)
Mean years since lymphedema began (range)	14 (2–40)	9.82 (3–21)	16.82 (4–33)	17.59 (2–40)	25.9 (1–49)	25.90 (1–49)	25.59 (1–45)	25.97 (1–49)
Mean years living in the endemic area (range)	47.70 (19–65)	45.96 (26–65)	47.53 (26–62)	50 (19–64)	41.7 (3–65)	42.17 (3–65)	42.27 (3–65)	41.81 (3–65)
Median MDA rounds (range)	5 (1–15)	5 (1–13)	5 (1–10)	5.5 (2–15)	4 (0–15)	4 (0–15)	4 (0–15)	4 (0–15)
History of ADL attacks, *n* (%)	59 (98.3)	24 (96)	15 (100)	20 (100)	82 (97.6)	23 (95.8)	35 (97.2)	24 (100)
Weight, median (range)	62.75 (42.5–90)	58.5 (42.5–90)	65.5 (53.5–82.5)	64.25 (48–77)	59.35 (40–125)	62.2 (40–94.5)	56.1 (42–100)	61.4 (43–125)
Occupation, *n* (%)								
Farmer	40 (66.7)	19 (76)	11 (73.3)	10 (50)	50 (59.5)	14 (58.3)	22 (61.1)	14 (58.3)
Trader	14 (23.3)	4 (16)	3 (20)	7 (35)	NA	NA	NA	NA
Other	6 (10)	2 (8)	1 (6.7)	3 (15)	34 (40.5)	10 (41.7)	14 (38.9)	10 (41.7)

**Table 3 pathogens-12-00809-t003:** Association with frequencies of CD4^+^ T cells in the study cohort. Uni- and multi-variable linear regression analyses were performed to adjust for all potential confounders (site, age, sex, years living in filarial endemic areas, years living with lymphedema, rounds of MDA received, and stage of lymphedema). Univariable analysis was carried out after adjusting for a site effect.

		Univariable			Multivariable		
Covariate	*n*	Coef.	95% CI	*p*-Value	Coef.	95% CI	*p*-Value
Site							
Ghana *	60	0.00	-	-	0.00	-	-
Tanzania	84	−8.83	−11.88 to −5.79	**<0.001**	−5.10	−10.00 to 0.06	0.05
Age (per year)	-	−0.01	−0.16 to 0.13	0.87	−0.01	−0.39 to 0.37	>0.90
Sex							
female *	102	0.00	-	-	0.00	-	-
male	42	−5.59	−8.97 to −2.21	**0.001**	−0.81	−6.90 to 5.20	0.80
Years living in the endemic area	-	−0.04	−0.15 to 0.07	0.50	−0.05	−0.34 to 0.23	0.70
Years with lymphedema	-	−0.14	−0.29 to 0.02	0.07	−0.14	−0.35 to 0.07	0.20
Rounds of MDA received	-	0.17	−0.33 to 0.67	0.51	0.26	−0.32 to 0.84	0.40
Stage grouping							
Stages 1–2 *	49	0.00	-	-	0.00	-	-
Stages 3–4	51	−2.43	−6.04 to 1.19	0.19	−2.10	−7.30 to 3.10	0.40
Stages 5–7	44	−4.70	−8.39 to −1.00	**0.01**	−5.00	−10.00 to −0.35	0.067

*n* = number of observations; Coef. = coefficient; 95% CI = 95% confidence interval; * reference stratum.

**Table 4 pathogens-12-00809-t004:** Association with frequencies of CD4^+^HLA-DR^+^CD38^+^ T cells in the study cohort. Uni- and multi-variable linear regression analyses were performed to adjust for all potential confounders (site, age, sex, years living in filarial endemic areas, years living with lymphedema, rounds of MDA received, and stage of lymphedema). Univariable analysis was performed after adjusting for a site effect.

		Univariable			Multivariable		
Covariate	*n*	Coef.	95% CI	*p*-Value	Coef.	95% CI	*p*-Value
Site							
Ghana *	60	0.00	-	-	0.00	-	-
Tanzania	84	4.28	3.14 to 5.41	**<0.001**	3.8	2.10 to 5.60	**<0.001**
Age (per year)	-	0.08	0.03 to 0.14	**0.002**	0.02	−0.11 to 0.15	0.80
Sex							
Female *	102	0.00	-	-	0.00	-	-
Male	42	0.13	−1.17 to 1.44	0.84	0.18	−1.90 to 2.30	0.90
Years living in the endemic area	-	0.04	−0.001 to 0.8	0.06	0.06	−0.04 to 0.16	0.20
Years with lymphedema	-	0.02	−0.04 to 0.08	0.49	0.00	−0.07 to 0.07	>0.90
Rounds of MDA received	-	−0.06	−0.26 to 0.13	0.51	−0.16	−0.36 to 0.04	0.12
Stage grouping							
Stages 1–2 *	49	0.00	-	-	0.00	-	-
Stages 3–4	51	−0.10	−1.41 to 1.21	0.89	−0.01	−1.80 to 1.80	>0.9
Stages 5–7	44	2.23	0.89 to 3.57	**0.001**	3.30	1.40 to 5.10	**<0.001**

*n* = number of observations; Coef. = coefficient; 95% CI = 95% confidence interval; * reference stratum.

**Table 5 pathogens-12-00809-t005:** Association with frequencies of CD4^+^CCR5^+^ T cells in the study cohort. Uni- and multi-variable linear regression analyses were performed to adjust for all potential confounders (site, age, sex, years living in filarial endemic areas, years living with lymphedema, rounds of MDA received, and stage of lymphedema). Univariable analysis was performed after adjusting for a site effect.

		Univariable			Multivariable		
Covariate	*n*	Coef.	95% CI	*p*-Value	Coef.	95% CI	*p*-Value
Site							
Ghana *	45	0.00	-	-	0.00	-	-
Tanzania	70	0.01	−0.02 to 0.05	0.49	−0.03	−0.10 to 0.04	0.40
Age (per year)	-	0.00	0.00 to 0.003	0.26	0.00	−0.01 to 0.004	0.71
Sex							
Female *	81	0.00	-	-	0.00	-	-
Male	34	−0.03	−0.07 to 0.11	0.15	−0.02	−0.10 to 0.06	0.57
Years living in the endemic area	-	0.00	−0.001 to 0.002	0.67	0.00	−0.004 to 0.003	0.67
Years with lymphedema	-	0.002	−0.484 to 0.004	0.06	0.003	0.00 to 0.01	**0.03**
Rounds of MDA received	-	0.00	−0.01 to 0.01	0.92	−0.001	−0.01 to 0.01	0.69
Stage grouping							
Stages 1–2 *	39	0.00	-	-	0.00	-	-
Stages 3–4	39	0.01	−0.03 to 0.06	0.54	−0.02	−0.09 to 0.04	0.49
Stages 5–7	37	0.39	−0.01 to 0.08	0.08	0.06	−0.01 to 0.12	0.09

*n* = number of observations; Coef. = coefficient; 95% CI = 95% confidence interval; * reference stratum.

**Table 6 pathogens-12-00809-t006:** Association with frequencies of CD8^+^PD-1^+^ T cells in the study cohort. Uni- and multivariable linear regression analyses were performed to adjust for all potential confounders (site, age, sex, years living in filarial endemic areas, years living with lymphedema, rounds of MDA received, and stage of lymphedema). Univariable analysis was performed after adjusting for a site effect.

		Univariable			Multivariable		
Covariate	*n*	Coef.	95% CI	*p*-Value	Coef.	95% CI	*p*-Value
Site							
Ghana *	45	0.00	-	-	0.00	-	-
Tanzania	65	0.08	−0.07 to 0.003	**<0.001**	−0.04	−0.09 to 0.02	0.15
Age (per year)	-	0.00	−0.003 to 0.001	0.39	−0.002	−0.01 to 0.002	0.28
Sex							
Female *	80	0.00	-	-	0.00	-	-
Male	30	0.02	−0.03 to 0.06	0.40	−0.004	−0.07 to 0.06	0.88
Years living in the endemic area	-	0.00	−0.001 to 0.001	0.96	0.001	−0.02 to 0.004	0.42
Years with lymphedema	-	0.00	0.00 to 0.003	0.13	0.00	−0.001 to 0.003	0.43
Rounds of MDA received	-	0.00	−0.01 to 0.002	0.20	−0.003	−0.008 to 0.003	0.36
Stage grouping							
Stages 1–2 *	38	0.00	-	-	0.00	-	-
Stages 3–4	38	0.01	−0.04 to 0.05	0.81	0.02	−0.03 to 0.07	0.48
Stages 5–7	34	0.06	0.02 to 0.11	**0.01**	0.05	−0.008 to 0.10	0.09

*n* = number of observations; Coef. = coefficient; 95% CI = 95% confidence interval; * reference stratum.

## Data Availability

The data presented in this study are available on request from the corresponding author.

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
