# Peer review of "Stage-Dependent Increase of Systemic Immune Activation and CCR5+CD4+ T Cells in Filarial Driven Lymphedema in Ghana and Tanzania"

_pathogens, 2023, doi:10.3390/pathogens12060809_

Round 1

Reviewer 1 Report

The study is interesting but not very novel and has the following limitations:

1. No hematology data including WBC and lymphocyte counts are presented. Since the whole study is based on frequencies of T cells, T cell counts are absolutely necessary for interpretation of data. Differences in frequencies in T cell subsets could merely be a reflection of differences in WBC or lymphocyte counts.

2. Additional clinical and demographic data can be added to the Tables.

3.   Kindly mention, how the authors excluded the other parasitic infection or any other co-morbidity details. Also, include the exclusion criteria.

4.  Authors could also include the systemic cytokine responses and show the relationship between the subsets.

5. Not sure focusing only on CCR5 and PD-1 alone in T cells is sufficient. Other markers can be examined.

·     

Minor language editing needed.

Reviewer 2 Report

The manuscript entitled "Stage dependent increase of systemic immune activation and CCR5+CD4+ T cells in filarial driven lymphedema in Ghana and Tanzania" Title, abstract and overall rationale of work is well written. However, there are still some minor concerns, which needs to be addressed before publication

1) Introduction section is written well and line number 72 author must be add references to justify the sentence.

2) Material method section: In the section 2.1 author must be write blood sample collection time like night or day (2.1. Study population and parasitic assessment).

3) During flow cytometry author used isotype control to make perfect gating instead of control group.

4) In material method section author need to incorporate how many sample they proceed and number of participant.

4) Results section: Figure 1a and 1b there are no any p-value is written and I recommend author to write there. However, in the legend author need to explain what is the figure 1a, b and c.

5) The quality of the figure 2, 4 and 5 are not good author need to improve the quality. However, author must be add statistical value of each graph to show the significance or non-significance value.

6) This study is preliminary and author need to check also cytokines and they only check phenotype of these cell during filarial infection in different stages and two countries. I suggest author they need to perform and check the status of pro-inflammatory and anti-inflammatory cytokines during infection.

7) Discussion is written well and compared this study to other study. However, author discussed to much details and compared this data to other parasites like Leishmania and other. I suggest author to reduce the discussion and write concise way.

8) Conclusion section must be separate and author also need to write limitation of this study.

9) Reference no 1 author need to provide details. Some references are too old for example references no 4, 6 and others. I suggest author to revise if other latest manuscript is available in the same information.

The English quality of this manuscript is good and written very clearly.

Round 2

Reviewer 2 Report

The authors have addressed all the concerns raised in the previous version of the manuscript and the quality has much improved after incorporating required modifications. Therefore, the manuscript may be considered for publication in this Journal.

The authors have addressed all the concerns raised in the previous version of the manuscript and the quality has much improved after incorporating required modifications. Therefore, the manuscript may be considered for publication in this Journal.